# City Health Examination and Evaluation of Territory Spatial Planning for SDG11 in China: A Case Study of Xining City in Qinghai Province

**DOI:** 10.3390/ijerph20043243

**Published:** 2023-02-13

**Authors:** Xiangjuan Zhao, Hanxuan Zhang, Jun Ren, Jing Guo, Quanxi Wang, Chengying Li

**Affiliations:** 1School of Civil Engineering, Qinghai University, Xining 810016, China; 2Graduate School of Qinghai University, Qinghai University, Xining 810016, China; 3Institute of Eco-Environment, Qinghai Academy of Social Sciences, Xining 810016, China; 4School of Grammar, Northeastern University, Shenyang 110169, China

**Keywords:** SDG11, territorial spatial planning, city health examination and evaluation, improved TOPSIS method, Xining city

## Abstract

City health examination and evaluation of territorial spatial planning is a new policy tool in China. However, research on city health examination and evaluation of territorial spatial planning is still in the exploratory stage in China. Guided by sustainable cities and communities (SDG11), a reasonable city health examination and evaluation index system for Xining City in Qinghai Province is constructed in this paper. The improved technique for order preference by similarity to ideal solution (TOPSIS) was used to quantify the evaluation results, and the city health index was visualized using the city health examination signals and warning panel. The results show that the city health index of Xining City continuously rose from 35.76 in 2018 to 69.76 in 2020. However, it is still necessary to address the shortcomings in innovation, coordination, openness and sharing and to improve the level of city space governance in a holistic way. This study is an exploration of the methodology used in city health examination and the evaluation of territorial spatial planning in China, which can provide a foundation for the sustainable development of Xining City and also provide a case reference for other cities seeking to carry out city health examinations and evaluations of territorial spatial planning in China.

## 1. Introduction

As large and complex organic systems [1,2], cities are not only an important carrier of human social and economic development, but also the “main battlefield” for building a beautiful China [3]. Whether cities are healthy and sustainable has an important impact on the improvement of human well-being and high-quality social and economic development [1,4]. Sustainable cities are the focus of many disciplines [5,6].Thus, territorial spatial planning control is important to optimize the interaction between the natural environment and land use and to resolve the conflict between social and economic development and ecological protection [7]. In 2015, member states of the United Nations committed to implementing 17 Sustainable Development Goals (SDGs) as part of the 2030 Agenda for Sustainable Development, and Sustainable Cities and Communities (SDG11) is central to achieving all SDGs [8,9,10,11]. China has observed tremendous levels of urbanization, from less than 20% in the 1980s to nearly 60% of the population today [12], and the urbanization rate is expected to reach 75% by 2030 according to the China Urbanization 2.0 report [13]. In “Major Issues in the National Strategy for Medium- and Long-term Economic and Social Development”, Chairman Xi put forward that “We should better promote people-centered urbanization, make cities healthier, safer and more livable, and become a space for people to live a high quality of life” [14]. Therefore, the study of sustainable cities has great relevance for building a beautiful China and realizing the United Nations Sustainable Development Goals. At present, international research on sustainable cities mainly focuses on climate change, urban governance and smart cities [6], in which urban governance is the key point. City health evaluation is the organic combination of direction- and goal-oriented urban means of assessment and has ecological, humanistic, high-quality and sustainable development characteristics [15,16]. The city health evaluation is an important means to promote modernization and the urban space management level, as well as an important measure for establishing a national spatial planning system and supervising the implementation of territorial spatial planning in China.

Urban living and health issues have become a focus of academics and policy-makers alike. The increase in urban construction land has occupied green areas within cities. Meanwhile, this reduction in green space has a negative effect on all people’s physical and mental health [17,18,19] Addressing the ecological consequences and health hazards caused by rapid urbanization has become a widely discussed issue [20]. The complex impact of the pandemic and how this extreme event has influenced urban sustainability remains to be understood [21]. Internationally, metropolitan cities such as London, Tokyo and Sydney have already entered the stage of construction, maintenance and planning review [22,23]. They have established relatively complete information systems, as well as planning, implementation and evaluation mechanisms, accumulating rich practical experience in planning and evaluation. City-level SDG assessments are beneficial to understand the strengths and weaknesses of the city for local governments and decision makers [24]. At present, the latest literature emphasizes the application of SDGs at the city scale, including case studies, and addresses methods and challenges associated with measuring SDG progress in cities—more specifically, for SDG 11 indicators [25,26,27]. City health examination and evaluation is a new model and policy tool for territorial spatial planning to evaluate implementation effectiveness in China [28]. In 2011, Shenzhen City first put forward the concept of a “city health examination” in the Construction of Urban Development Assessment System and 2011 Annual City Development Assessment. At the Central City Conference in 2015, Chairman Xi proposed the requirement of “establishing a city health examination and evaluation mechanism”. Meanwhile, “The Notice on the Construction of “one map” and the Status Evaluation of National Spatial Planning” was issued, which specified all localities in which to carry out the health examination and evaluation of national spatial planning [29,30], marking the official beginning of the physical examination and evaluation of territorial spatial planning. In 2020, the Land and Space Planning Bureau of the Ministry of Natural Resources issued “The Notice on Carrying out the City Healthy Examination and Evaluation of the Current State Review of Urban Territorial Space Planning”, aiming to further explore and improve the mechanism and experience of city health examination and evaluation of territorial spatial planning. In May 2021, the Ministry of Natural Resources issued “Regulations for Healthy Examination and Evaluation of Territory Spatial Planning” (TD/T1063-2021, referred to as the “Regulations”), which constructed the evaluation index system and put forward clear requirements for results, aiming to establish and supervise the implementation of the national territorial spatial planning system so that China could speed up the establishment of a healthy examination and evaluation system for territorial spatial planning that is conducted every year and every five years.

City health examination and evaluation of territorial spatial planning is an important part of the refinement and full-cycle management of territorial space and an important tool through which to promote high-quality urban development and improve the effectiveness of the implementation of territorial spatial planning. However, with the prominence of the public policy attribute of territorial spatial planning [31,32,33], the city health examination and evaluation of territorial spatial planning have become more systematic and complicated. Although the “Regulations” put forward a specific index system and achievement requirements for city health examination and evaluation, the evaluation results were mainly qualitative conclusions, which could not directly quantify the health examination results and allow vertical and horizontal comparisons, and the guidance for the implementation of territorial space planning and dynamic monitoring was still insufficient. In addition, challenges remain in downscaling targets and indicators to the city level to support planning and policy in a local context [25]. Meanwhile, unilateral implementation evaluation studies on urban master planning [17,18,34] and land use master planning [35,36,37] are relatively common in the academic community, but relevant studies on the theoretical basis and corresponding technical methods for city health examination and evaluation of territorial spatial planning in the new era of “integration of multiple plans” are still lacking [16,36]. Therefore, in terms of practical work and academic research, city health examination and the evaluation of territorial spatial planning are in their initial stages, and evaluation indicators, evaluation methods and application of evaluation results need to be further explored and improved in combination with local practices [16,38,39,40,41]. So, constructing a reasonable city health examination and evaluation index system of territorial spatial planning and quantifying and visualizing the evaluation results are very important. Here, we attempted to construct a reasonable city health examination and evaluation index system of territorial spatial planning, and the improved TOPSIS method was applied to quantify and visualize the evaluation results for better understanding of sustainable cities and communities.

In view of this need, with the guidance of the United Nations 2030 Sustainable Development Goals (SDGs), this paper takes Xining City as a study case and constructs a set of index systems to evaluate the characteristics of Xining. The improved TOPSIS method was applied to construct the signal light and early warning board of city health examination in an exploratory way and to quantitatively evaluate and visually express the results of the health examination and the evaluation of Xining’s territory spatial planning from 2018 to 2020 through the city health index. In summary, this paper is an exploration of the quantitative research methodology for city health examination and an evaluation to provide methodology for the spatial governance and high-quality development of Xining. It also provides a case reference for other cities seeking to carry out health examinations and evaluations of territorial spatial planning in China.

The rest of this paper is organized as follows. Section 2 introduces the study area and data sources, which includes the social statistics data and natural resource data. It also introduces the index system and research methods in detail. As the core section, Section 3 presents the results and analysis, in which we analyzed the characteristics of the six dimensions of the city health index. Section 4 provides a discussion comprised of the rationality of the improved TOPSIS method, suggestions on city sustainable development and community construction, limitations and prospects. The last section presents our conclusions.

## 2. Study Area and Methodology

### 2.1. Study Area

The geographical coordinates of Xining City are 100°52′04″~101°54′54″ E, 36°13′42″~37°28′11″ N, which is located in the Huangshui Valley at the eastern foot of Riyue Mountain (shown in Figure 1). The landforms of Xining are complex and diverse, with an average elevation of 2275 m and a semiarid continental plateau climate. Xining is the capital of Qinghai Province, with an area of 7606.78 km^2^. In 2020, the GDP of Xining City was 137.298 billion yuan, the permanent resident population was 2.468 million, the urbanization rate was 78.63% and the ethnic minority population accounted for 28.57% of the permanent resident population. With 1.1% of the whole province’s land, Xining accounted for 41.6% of the population and generated approximately 62% of the GDP for Qinghai Province. Xining is the only central city on the Qinghai-Tibet Plateau with a population of more than one million, and it is also one of the important node cities for both the “One Belt and One Road” development strategy and the upper reaches of the Yellow River Basin. Because the highland valley urban landform and the tributaries of the Yellow River divide the city and restrict the development of the fragile ecological environment, the land use in Xining City is approximately 90% agricultural land, with 0.5% construction land and 70% forest and grassland with 30% cultivated land. With a good ecology, livable environment and strong sense of well-being, Xining has won the titles of National Civilized City and National Forest City in China, as well as becoming the first provincial capital to be selected as the pilot “No Waste City”. However, there also exist some problems, such as shortages of land resources, the balance of farmland and the allocation of public service facilities, suggesting that urban sustainable development needs to be explored. Therefore, it is typical and representative to select Xining City as the study case for the city health examination and evaluation of territorial spatial planning.

### 2.2. Data Sources

The social statistics data were mainly obtained by *Statistical Yearbook of Xining City* and the Xining Natural Resources and Planning Bureau from relevant commissions and bureaus through correspondence. The natural resource data were mainly obtained through the third National Land survey data of Xining City, the annual land survey change data of Xining City in 2020, and geographical situation monitoring and high-resolution remote sensing images, which also integrate spatiotemporal big data such as the POI data of Xining City, big data on city operation, mobile phone signaling data and night light data. The ecological and environmental quality data were obtained from the Xining Municipal Ecological and Environmental Quality Bulletin, and the water resources data were obtained from the Qinghai Provincial Water Resources Bulletin. At the same time, various channels, such as social satisfaction surveys [42], were adopted to obtain demands and suggestions on living environment, people’s well-being and city conditions of local residents.

### 2.3. Construction of the Index System

As the basis for the city health examination and evaluation of territorial spatial planning, the design of the index system should take the efficient and sustainable utilization of national space as the starting point and take into account not only vertical continuity, horizontal universality and comparability, but also regional heterogeneity [15,16]. Meanwhile, it should adhere to the principle of combining subjectivity and objectivity, which can not only reflect the objective patterns of city development but also meet the subjective demands of individual residents [38]. This approach can finally form a research method system in which objective evaluation and subjective evaluation verify and complete each other to better reflect the development concept of a “people-oriented” city and a “people’s city for the people”.

This paper constructed the index system from the following three aspects. First, according to the “Regulations”, the evaluation index system is composed of 33 basic indicators and 89 recommended indicators. Xining is a city approved by the State Council, and 48 indicators marked “▲” in the recommended indicators are mandatory indicators for cities approved by the State Council, but Xining is an inland city, and 5 indicators of ocean (B38, 39, 40, 65) and rail transit (B70) are not involved. At the same time, the construction of Beautiful China is the specific practice of the SDGs in China, and territorial spatial planning is an important carrier of the construction of Beautiful China [41,42]. Referring to the “Beautiful China Construction Evaluation Index System and Implementation Plan” and “Guidelines for the Formulation of Municipal Territorial Space Master Plan (Trial)”, we modified “local indicator species” to “conservation rate of key species”, “coverage area of the distribution of clean energy facilities” to “proportion of new energy and renewable energy” and “total city building volume” to “national land development intensity”, which are important indicators and are difficult to obtain. This method not only facilitates the acquisition of data and improves the credibility of the indicators, but also effectively links the formulation of territorial spatial planning with the building of a beautiful China. Second, to fully guarantee residents’ right to participation, knowledge and supervision in the city health examination and evaluation of territorial spatial planning [39], six social satisfaction survey indicators were added to better convey the perceptual demands of residents. Finally, combined with the characteristics of Xining City, e.g., building a model city of green development, building a resilient city and ethnic unity and progress, we added 12 local characteristics indicators, such as city street neatness rate, average annual concentration of fine particulate matter (PM2.5) and safe utilization rate of polluted farmland. Through the process mentioned above, the city health examination and evaluation of the territory spatial planning index system of Xining was constructed with “79 basic indicators (A, B▲) + 38 recommended indicators (B) + 6 social satisfaction indicators (S) + 12 local characteristics indicators (T)” (shown in Table 1).

To ensure the scientific rigor of the index weight, the weight of each index was obtained using the entropy method, and the weights of the criterion layer and target layer were obtained via the combination of the entropy method and an expert scoring method [39,43].

### 2.4. Methodology

#### 2.4.1. Maximum–Minimum Method

The maximum–minimum values were used as the standard treatment for each index. The specific operation process is as follows:

According to the upper limit $\;(X_{imax}$) and the lower limit $\left(X_{imin}\right)$ thresholds of each indicator, the extreme value method was adopted to conduct dimensionless standardization of each indicator. The calculation formula is as follows:

For positive indicators:(1)$$x_{ij}=\frac{X_{ij}-X_{imin}}{X_{imax}-X_{imin}}$$

For negative indicators:(2)$$x_{ij}=\frac{X_{imax}-X_{ij}}{X_{imax}-X_{imin}}$$

In Equations (1) and (2), $x_{ij}$ is the standardized value of the extreme value method of the *i*th index in the *j*th year; $X_{imax}$ and $\;X_{imin}$ are the maximum and minimum values of the *i*th index, respectively. Since the target value or optimal threshold of each indicator is not given in the Regulations, the ideal value obtained from various specifications, documents or plans is taken as the maximum value in this study.

#### 2.4.2. Improved TOPSIS

The TOPSIS method is a multiobjective decision analysis method based on the proximity between the evaluation object and the ideal scheme [44]. The improved TOPSIS method is a multischeme decision-making method combined with the entropy method and the traditional TOPSIS method, which has the advantages of no strict restrictions on the data distribution and the number of indicators, simple calculation and longitudinal comparison of the evaluation results [45]. The specific operation process is as follows:

① The normalization matrix is constructed to obtain the dimensionless decision matrix
(3)$$Z=(z_{ij})m\times n\;\;Z_{ij}=\frac{x_{ij}}{\sqrt{\sum_{i=1}^{n}x_{ij}^{2}}}\;i=1,\ldots ,n;\;j=2018,\ldots ,m\;$$

In Equation (3), $Z_{ij}$ is the normalized value after dimensionless normalization, and $x_{ij}$ is the normalized value of the extreme value method. This study takes Xining City of Qinghai Province as the evaluation unit and selects 135 indicators in 2020 to evaluate the physical examination of Xining’s territorial space planning, so *m* = 2020, *n* = 135.

② Construct the normalized weighted decision matrix.
(4)$$Z^{\prime }=\left(z_{ij}^{\prime }\right)\;m\times n\;\;Z_{ij}^{\prime }=w_{i}\times z_{ij}\;i=1,\ldots ,n;\;j=2018,\ldots ,m$$

In Equation (4), $w_{i}$ is the weight value of the ith index. $z_{ij}$ is the normalized value after non-dimensionalization.

③ Determine the positive and negative ideal value schemes for $Z^{+}$ and $Z^{-}$.
(5)$$\left\{\begin{array}{c} Z^{+}=\left(Z_{1}^{+},Z_{2}^{+},\ldots ,Z_{n}^{+}\right)=\left\{maxZ_{ij}^{\prime }\left|i=1,\ldots ,n;j=2018,\ldots ,m\right.\right\}\\ Z^{-}=\left(Z_{1}^{-},Z_{2}^{-},\ldots ,Z_{n}^{-}\right)=\left\{maxZ_{ij}^{\prime }\left|i=1,\ldots ,n;j=2018,\ldots ,m\right.\right\} \end{array}\right.$$

In Equation (5), $Z^{+}$ is the maximum set of the *i*th index in the *j*th year; $Z^{-}$ is the set of minimum values in the *j*th year of the *i*th index.

④ Using weighted Euclidean distance measure distance $D_{j}$.
(6)$$\left\{\begin{array}{c} D_{j}^{+}=\sqrt{\overset{n}{\underset{i=1}{\sum }}{\left(Z_{ij}^{\prime }-Z_{i}^{+}\right)}^{2}}\\ D_{j}^{-}=\sqrt{\overset{n}{\underset{i=1}{\sum }}{\left(Z_{ij}^{\prime }-Z_{i}^{-}\right)}^{2}} \end{array}\right.$$

In Equation (6), $D_{j}^{+}$ and $D_{j}^{-}$ are the positive ideal value distance and negative ideal value distance of the *j*th year, respectively.

⑤ Measure criterion layer city health index value $H_{j}$.
(7)$$\;H_{j}=W_{i}*\frac{D_{j}^{-}}{D_{j}^{-}+D_{j}^{+}}$$

In Equation (7), $H_{j}$ is the urban health index value of the *j*th year criterion layer; $W_{i}$ is the weight of the criterion layer.

⑥ Measure the total value of the urban health index.
(8)$$\left\{\begin{array}{c} H_{J}=\sum W_{I}H_{j}\\ H=\sum H_{J} \end{array}\right.$$

In Equation (8), $H_{J}$ and *H* are the value and total value of urban health index of sub-target layer in the *j*th year, respectively. $W_{I}$ is the weight of the sub-target layer. When *H* = 1, the city health state is the best; vice versa, the city health state is the worst.

#### 2.4.3. Construction of Signal Lights and Warning Boards for City Health Examination

Based on the evaluation concept of China’s provincial sustainable development goals by Wang et al. [8] and the traffic light system developed by the International Union for Conservation of Nature (IUCN) [46,47,48] for matching degree division of nature-based solutions, we explored the construction of city health examination signal lights and early warning signals for territorial spatial planning (shown in Figure 2). The red signal light indicates a state of serious alarm, which suggests that the health of the city is seriously threatened and corresponding measures should be taken to improve it. The orange signal light indicates a state of medium alarm, which means the city is in a stage of poor health and still needs to take necessary measures to make up for the shortcomings. The yellow signal light indicates a light alarm state, which implies that although the city can operate normally, the city indicators still need to be consolidated and improved. The green signal light indicates that there is no alarm state, and all indicators of the city are in the optimal cooperative combination state, which can be maintained continuously. The assessment results can be visualized on the early warning board of the city health examination, which can be used to intuitively evaluate the health status of the city. In addition, the results of the city health examination and evaluation are ranked, and the change trend of the city health examination score is indicated.

## 3. Results and Analysis

### 3.1. Analysis of the City Health Index

As shown in Figure 3, the total city health index of Xining City continued to rise during the study period, from 35.76 in 2018 to 69.76 in 2020, an increase of 49.17% compared to the level in 2018. Overall, the city health status of Xining City continues to improve, but the overall city health index is still below the high-value area. The COVID-19 epidemic especially has had a negative impact on the openness of Xining City. Thus, it is still necessary to improve the city health level and comprehensively improve the state of the city’s health.

### 3.2. Analysis of Six Dimensions of the City Health Index

#### 3.2.1. Overall Situation of the Six Dimensions of the City Health Index

As shown in Figure 4, the city health index of security in Xining showed a U-shaped change, decreasing from 51.08 in 2018 to 40.42 in 2019 and then increasing to 66.62 in 2020. The city health index of innovation increased continuously in a “straight line” pattern, from 43.36 in 2018 to 85.99 in 2020. The city health index of coordination showed a U-shaped change, decreasing from 52.64 in 2018 to 29.62 in 2019 and then increasing to 48.09 in 2020. The green city health index increased continuously in a “straight line” pattern, from 21.79 in 2018 to 76.21 in 2020. The city health index of openness increased continuously in a “straight line” pattern, from 31.09 in 2018 to 58.98 in 2020. The city health index of sharing increased continuously in a “straight line” pattern, from 16.97 in 2018 to 79.41 in 2020. In 2020, the city health index of innovation, greenness and sharing in Xining City were in the high area, the city health index of security and openness were in the median area and the city health index of coordination was in the low area.

As seen from Figure 5, Xining City’s state of alarm in the dimensions of innovation, greenness and sharing continued to weaken, and the ranking of the evaluation results increased as a whole, showing a trend of slow, continuous rising and stable change. In 2020, Xining City was in a state of no alarm. The security dimension of alarm increased first and then weakened, and the overall ranking of the evaluation results decreased, showing a fluctuating trend of declining rapidly first and then rapidly rising. It was in a state of light alarm in 2020. The alarm state of the coordination dimension increased first and then weakened, and the ranking of the evaluation results decreased overall, showing a fluctuating trend of declining rapidly and then slowly rising. It was in a state of light alarm in 2020. The alarm state of the openness dimension continued to weaken, and the ranking of the evaluation results rose as a whole, showing a continuous and slow upward trend. It was in a state of light alarm in 2020. In short, the six dimensions of Xining’s security situation were all weakening, while the results in innovation and greenness are remarkable. Effective measures should still be taken in security, coordination, openness and sharing to build a smarter, more inclusive and more open city, as well as comprehensively improving the level of urban public services and people’s well-being.

#### 3.2.2. Security Dimension

Figure 6 shows that Xining City has achieved remarkable results in water security, ecological security, city resilience, planning control and city security satisfaction. As an important central city in the upper reaches of the Yellow River Basin, Xining has firmly established a sense of the bottom line, given priority to delineating the red line of ecological protection, implemented the strictest water resources management system, comprehensively established a four-level river and lake chiefs system, and successfully become a national water ecological civilization city. Xining City attaches great importance to planning and control, strictly enforces the implementation and requirements of planning and control and vigorously carries out the construction of city comprehensive pipe corridors and “sponge cities”. In 2020, the security utilization rate of polluted farmland in Xining City reached 98.00%, the renewal rate of the old pipe network reached 12.30% and the satisfaction rate of city security reached 93.64. In terms of water security, food security, ecological security, city resilience, planning control and city security satisfaction, the alarm state continued to weaken and remained in the state of no alarm, and the overall ranking of the assessment results rose, showing a slow upward trend. However, the state of alarm of Xining City in food security and cultural security aspects still needs to be given more attention. Xining adheres to the red bottom line for food security, and valley terrain limits city development. City land is the city space surrounding the “bite”. At the same time, because of the planting structure adjustment and the implementation of the policy of returning farmland to forest and grass, Xining is gradually adjusting to its sensitive areas of farmland soil and water loss, all of which place the area within the scope of efforts to reduce the quantity of cultivated land. Although the alarm state of food security in Xining City continues to weaken and the ranking of the evaluation results has risen as a whole, Xining City will face increasing pressure to complete the task of balancing farmland occupation and compensation in the future, and the situation still needs to be given more attention. In terms of cultural security, historical and cultural blocks and historical buildings have been actively promoted in Xining City to determine limits to development, to prevent the destruction of culturally relevant sites and their environment and to weaken the trade in historical relics. The evaluation results indicated an overall rapidly rising trend in these metrics. Xining is a multiethnic city; however, its cultural diversity and public cultural identity are not high, and great importance still needs to be attached to industry.

#### 3.2.3. Innovation Dimension

As shown in Figure 7, Xining has achieved remarkable results in terms of smart cities. Xining City vigorously carried out the strategy of innovation-driven development to advance the construction of a data-driven wise city, and in strict accordance with the national spatial basic information platform construction and requirements of a picture, it preliminarily built the Xining national spatial planning-based information platform “picture”, implementing the unified platform and its application to provide complete coverage of units at the county level. Aspects of a smart city have always been in a state of no alarm in Xining, and the ranking results have always ranked these characteristics in first place, showing a stable trend of change. At the same time, Xining City’s input and output of city innovation, development mode and city innovation satisfaction have weakened. Xining clings to the strategy of building the “area” and the new era of opportunity. Xining has sought to follow the pattern of countries adhering to a Western development policy, focusing on five industrial clusters centered around lithium electricity, photovoltaic solar-thermal, nonferrous alloy characteristics of high-tech and new materials, the chemical industry, biological medicine and intensive processing of plateau flora and fauna resources. Xining is home to “ecological farming and animal husbandry in the state key laboratory” and the “state key laboratory of Tibetan medicine new drug development”, and it has sought to introduce and cultivate high-level innovative entrepreneurial talent. Xining has worked toward the implementation of innovative entrepreneurial support and the “lead to gather only 555 plan”, and it has striven to build and aggregate talent to achieve new breakthroughs. At the same time, Xining City has actively promoted production, set aside industrial park land for use as potential stock, adjusted and optimized its city development patterns and sought to improve public satisfaction with the degree of innovation in the city. In terms of city innovation input and output, development mode and city innovation satisfaction, the state of alarm has weakened, and the ranking of the evaluation results has increased rapidly as a whole.

#### 3.2.4. Coordination Dimension

As shown in Figure 8, the alarm state in terms of agglomeration, city–rural integration and satisfaction with city coordination in Xining City has decreased, while the alarm state in terms of aboveground and underground coordination has been aggravated. Beautiful towns in Qinghai Province have demonstrated a development plan that emphasizes their location on the plateau, and Xining City continues to be highlighted in this regard. This group of cities incorporates the plateau ecological landscape into their development patterns. Xining has pursued the role of the plateau in its efforts to develop into a beautiful town, promoting this aspect in its county districts and accelerating the development of city and rural integration. The permanent population urbanization level has reached 78.63%, and the ratio of city to rural residents’ income is 2.74:1. To further improve the equality of public services between city and rural areas and improve public satisfaction with city coordination in the area, the integration of city and rural areas and city industry can be improved. The evaluation results indicate an overall rapid increase in this area. However, among the districts of Xining in 2019, the conditions in Huangzhong had a strong affect on the city within the scope of indicators such as the per capita area of underground space, and the growth in the area of underground space was far less than that required by the population growth, resulting in a decline in the index indicator “bluff type” and, in turn, affecting the evaluation results for underground indicators. In terms of the alarm state of the underground environment, the earth as a whole decreased from no deterioration to a high alert, dropping this indicator to last place in the evaluation, showing a rapid downward trend.

#### 3.2.5. Green Dimension

As shown in Figure 9, Xining City has achieved remarkable results in ecological protection, green and low-carbon life and greenness satisfaction, but it still needs to pay great attention to the warning situation in greenness and low-carbon production. Guided by the philosophy that clean waters and lush mountains are just as valuable as gold and silver, Xining has carried out holistic conservation and systematic governance of its mountains, rivers, forests, farmlands, lakes, grasslands and deserts. The city has relied on the construction of a natural protected area system and has sought to serve as a demonstration of a city that is actively integrated into the national park system. Xining has sought to advance the greening of the plateau and advocated for blue, clear lakes through activities such as starting six large construction projects, promoting the new mode of “park city + natural protected area” and vigorously promoting the transformation and upgrading of industry. Xining has striven to build a modern industrial system, construct a state-of-the art waste management, reduction and recycling system, and improve city and rural residents’ awareness of energy conservation, emissions reduction and trash sorting. The development of Xining as a green city has become a basic tenet of its development, and happiness and beauty have become incorporated into the Xining lifestyle, with ecology providing a competitive advantage toward this aim in Xining. The rate of forest coverage in Xining reached 36.00% in 2020, new green buildings accounted for 62% of new construction, the green transportation ratio was above 65% and the rural garbage disposal rates were above 98%. The city was named a national forest city and national garden city in recognition of its ecological protection, greenness and promotion of low-carbon lifestyles. Residents’ satisfaction with life in the city has continued to improve, and the overall ranking of the evaluation results rose, showing a rapid trend of improvement. Although the green and low-carbon production of Xining City was in a light alarm state in 2020 and the ranking of the evaluation results decreased as a whole, the alarm state still needs to be given careful attention. With the continuous implementation of the national “double carbon” target, Qinghai Province is an important part of the national “three regions and four belts” ecological security barrier. Accelerating the formation of a low-carbon and green production mode is still the focus of the future industrial transformation and development of the province. As the first provincial capital city selected as the pilot of a “nonwaste city”, Xining City should vigorously promote the new energy industry and green and low-carbon production mode, accelerate the upgrading of the park system and improve the economic system of green and low-carbon circular development.

#### 3.2.6. Openness Dimensions

As shown in Figure 10, Xining’s alarm state in terms of satisfaction with network connectivity and city opening to the outside world has diminished, while the alarm state in terms of foreign communication and trade has been aggravated. Xining City is a remote northwestern inland city, and its location advantages of adjoining Tibet and Sichuan, linking Xinjiang and facing Lanzhou’s foreign economic and trade cooperation provide a good foundation for both domestic and international communication and trade. The “Tour de Qinghai Lake international road cycling race” and “national ecological tourist purpose” action in Qinghai Province were highly visible abroad. With the continuous upgrading of Xining Caojiapu International Airport, Xining’s network connectivity capability is enhanced, the city’s visibility is enhanced, the public’s satisfaction with the city’s opening to the outside world is continuously improved and the warning level is continuously decreased. The ranking of the evaluation results is also rising, showing a rapid trend of change. Xining is home to Xining Caojiapu International Airport, with regular international flights to 7 cities and domestic flights to 75 cities. Compared with other provincial capitals, foreign and domestic routes into and out of Xining are still very limited. In addition, affected by the COVID-19 epidemic, Xining’s domestic and foreign trade is depressed, as is its cultural and tourism industry. The number of inbound tourists in 2020 was 1/8 of that in 2018, and the total amount of foreign trade in 2020 was 5/9 of that in 2018. All these factors have led to an increased alarm state in Xining’s foreign exchanges and foreign trade, and the ranking of the evaluation results is also declining rapidly.

#### 3.2.7. Sharing Dimensions

Figure 11 shows that in the appropriate industry, the livability of the city is satisfactory. The alarm state on related indicators is declining, and the evaluation results are exhibiting a rapidly increasing change trend. These improvements are mainly due to Xining’s support for its board of education, health care and pensions and reductions in dust and noise pollution. Xining has vigorously carried out river improvements and ecological restoration of its territorial space, actively promoted garbage sorting and the use of green energy and, relying on the “smart city management” platform and the national campaign to create civilized cities, improved the city appearance and living environment. The rate of good air quality increased to 87.3 percent, and the per capita green area of public parks reached 13 square meters. Xining is among the top 10 happiest cities in China. However, Xining still needs to pay great attention to the alarm state in terms of city tourism and city sharing satisfaction. Due to the impact of COVID-19 and the withdrawal of Huangzhong County into districts on city indicators, the warning situation in 2020 is still in a state of light alarm, and the ranking of evaluation results is also declining, although overall, it still shows a trend of increasing rapidly.

## 4. Discussion and Conclusions

### 4.1. Discussion

Scholars have discussed the examination of city health from the perspectives of geography and city planning [15,16,40,41], proposed the basic principles of city health examination and evaluation and explored evaluation methods, but further exploration and improvement are needed [18,49]. This paper used the improved TOPSIS method to devise the “signal lights” and alarm states of indicators pertinent to a city health examination, as well as the results of territorial spatial planning in Xining City. Previous results were not quantified in length and breadth and were not expressed visually in the literature on city health examination. The present study provides an exploration of the methodology of city health examination and evaluation, which is beneficial to enriching the methodology, adding relevant cases and helping city managers improve city governance. However, this paper only studied one city. The indicators used provide key points with which to structure a high-precision research method for city health examination and evaluation and an evaluation index system for “city–community”, enabling a more detailed nested-target index of spatial indicators. In addition, determining the ideal values of these indicators is the key for dimensionless processing of the index, which still needs to be further refined and improved in future studies.

The results show that the city health index of Xining is still below the area with high values, and it still needs to be improved using comprehensive measures. There are six aspects of city health examination and evaluation that can achieve the goal of city sustainable development and community construction. First, security is an important precondition for building a livable and happy city. Second, technological innovation is the “invisible wing” of development. Third, coordination is the most important aspect that is beneficial to governing the city. In addition, openness and sharing are basic features. Finally, greenness is the ultimate goal of high-quality life. Xining has an unprecedented opportunity for development because there are many strategies available, such as “The Belt and Road initiative”, high-quality development, ecological protection of the Yellow River Basin and city agglomerations with Lanzhou and Xining. Therefore, Xining City should improve the spatial governance capacity of the city in the six aspects listed above. In terms of security, it should plan and determine “three zones and three lines”, innovate and improve a balanced system of the occupation and supplementation of cultivated land, ensure the baseline of food security and the red line for protecting farmland and actively delimit the protection of cultural relics and historical buildings to discover the value of cultural resources. For innovation, Xining should spend more on technological innovation, stimulate the vitality of innovation and build a more intelligent and convenient city. For coordination, it should consolidate and improve the achievements of rural revitalization and poverty alleviation, take the county town as the arena in which to build urbanization and boost city–rural integration and aboveground and underground development. For greenness, openness and sharing, Xining should continue its efforts to construct an ecological civilization; promote new infrastructure that is low-carbon, sustainable and resilient; strengthen its efforts at opening up and trade; improve its network of external connectivity; improve the city’s core competitiveness and build a happy city that benefits everyone.

### 4.2. Conclusions

In this paper, the improved TOPSIS method was used to quantify and visualize the results, and the signal lights and alarm states of city health examination and evaluation of territorial spatial planning were exploratively constructed from 2018 to 2020 in Xining City. The main conclusions are as follows:

(1) During the study period, the city health index of Xining continuously rose from 35.76 in 2018 to 69.76 in 2020, but it was still in the lower-value area; thus, the city still needs to greatly improve its health level. At the same time, the city health index of security and coordination changed in a “U”-shaped pattern, and the city health index of innovation, greenness, openness and sharing continued to rise in a “straight line”. Alarm states related to innovation, greenness and openness were declining throughout the study period. However, in terms of security and coordination, the alarm states increased at first and then decreased. Although Xining City has achieved remarkable success in innovation and greenness, the COVID-19 epidemic has had a negative impact on economic development and the urbanization process; it still needs to take targeted measures in the aspects of security, coordination, openness and sharing to complement the shortcomings and comprehensively improve the city’s spatial governance.

(2) Xining has achieved remarkable gains in fourteen aspects, including water security and ecological security. Alarm states related to innovation, greenness and openness continue to weaken and are in a state of no alarm. The results in eleven aspects, such as food security, cultural security and satisfaction with openness, were just passable. The alarm states were weakened but in a light state. The situation was more severe in three aspects, including overall planning of the aboveground and underground environment, foreign affairs and foreign trade, and the alarm states were sustained under a secondary state.

(3) Xining should actively participate in the construction of a better ecological civilization, consolidate and improve the achievements of rural revitalization and poverty alleviation, plan and determine “three zones and three lines”, increase investment in scientific and technological innovation, promote new infrastructure that is low-carbon, sustainable and resilient, comprehensively enhance the core competitiveness of the city and build a city of happiness that all people enjoy together.

## Figures and Tables

**Figure 1 ijerph-20-03243-f001:**
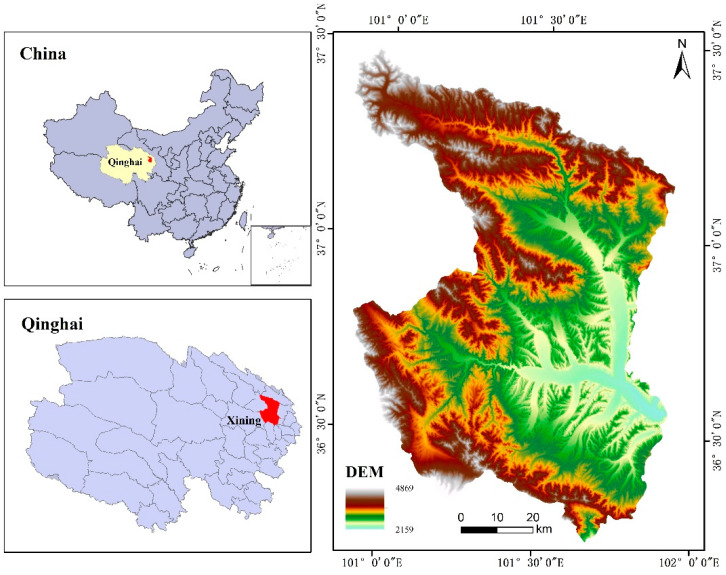
The location map of the study area (drawn by the authors).

**Figure 2 ijerph-20-03243-f002:**
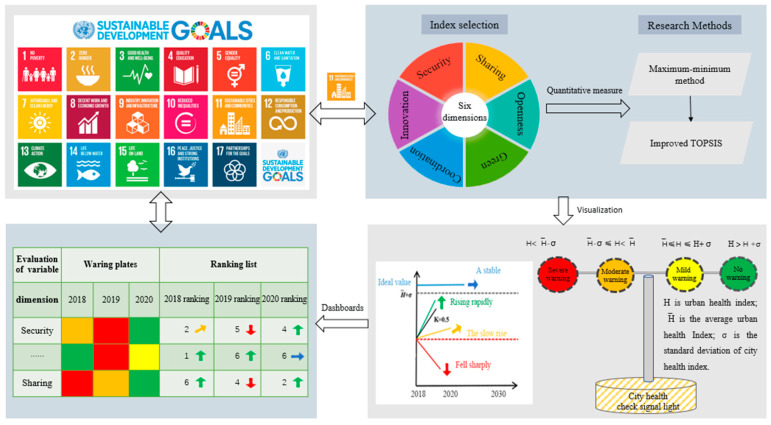
The construction process diagram of the city health examination signal light and warning panel in territorial spatial planning. (draw by authors).

**Figure 3 ijerph-20-03243-f003:**
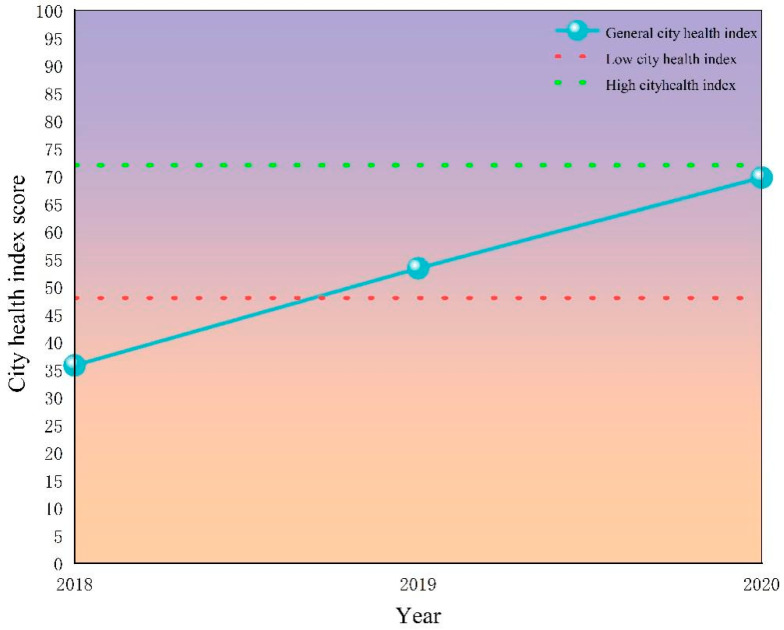
Changes in the total city health index in Xining City from 2018 to 2020.

**Figure 4 ijerph-20-03243-f004:**
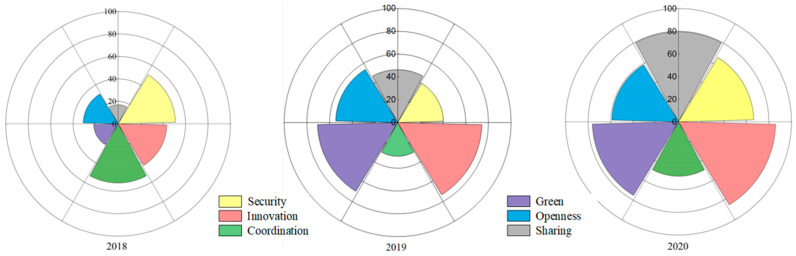
Changes in the six dimensions of the city health index in Xining from 2018 to 2020.

**Figure 5 ijerph-20-03243-f005:**
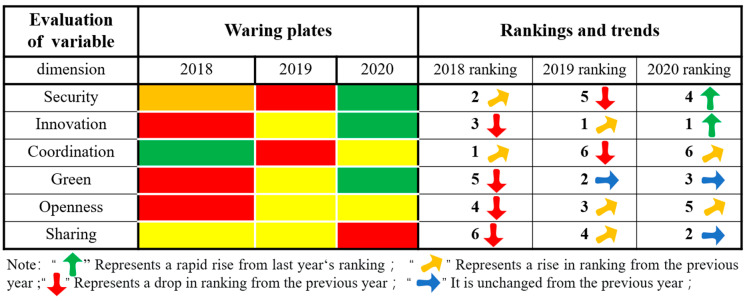
The results for six dimensions of city health examination and evaluation in Xining City.

**Figure 6 ijerph-20-03243-f006:**
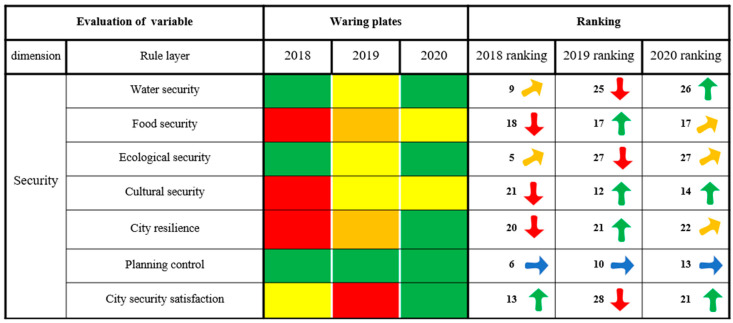
The results of the security dimension of city health examination and evaluation in Xining City.

**Figure 7 ijerph-20-03243-f007:**
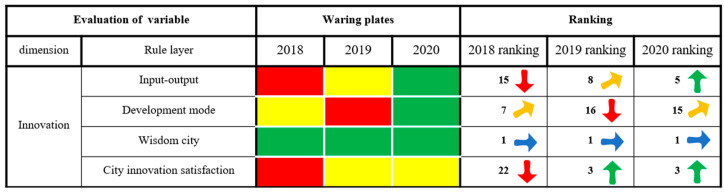
The results of the innovation dimension of city health examination and evaluation in Xining City.

**Figure 8 ijerph-20-03243-f008:**
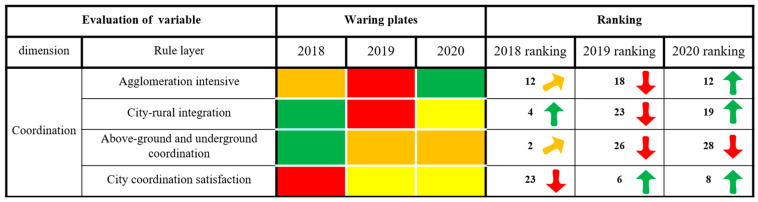
The results of the coordination dimension of city health examination and evaluation in Xining City.

**Figure 9 ijerph-20-03243-f009:**
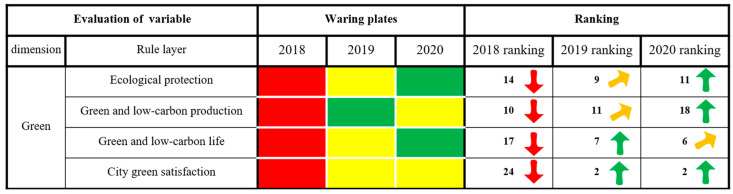
The results of the green city dimension of health examination and evaluation in Xining City.

**Figure 10 ijerph-20-03243-f010:**
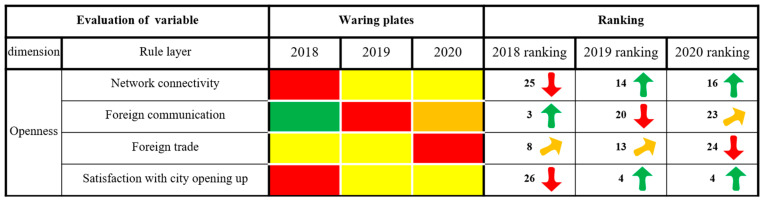
The results of the openness dimension of city health examination and evaluation in Xining City.

**Figure 11 ijerph-20-03243-f011:**
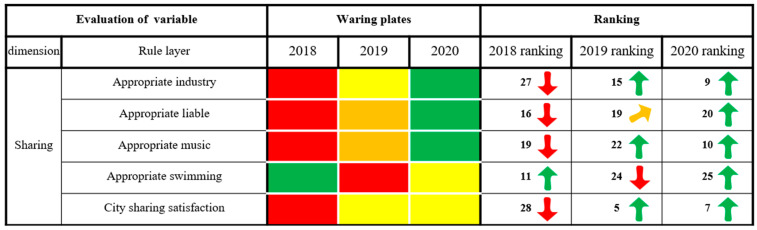
The results of the sharing dimension of city health examinations and evaluations in Xining City.

**Table 1 ijerph-20-03243-t001:** City health examination and evaluation index system of territorial spatial planning in Xining City.

Target Layer	Rule Layer	Index Layer
Security (0.18)	Water security (0.15)	A01 Per capita annual water consumption (0.0076); A02 Ground water level (0.0080); B01 Water quality of major rivers and lakes reaching the standard rate (0.0077); B02▲ Total water use (0.0076); B03▲ Water resources development utilization rate (0.0077); B04▲ Wetland area (0.0089); B05▲ River and lake water surface rate (0.0069); B06 Proportion of groundwater supply to total water supply (0.0070); B07▲ Recycled water utilization rate (0.0084); T01 Water quality compliance rate of city centralized drinking water source (0.0064)
	Food security (0.15)	A03 Protected area of permanent basic farmland (0.0064); A04 Cultivated land quantity (0.0078); B08 High-standard farmland area proportion (0.0070); T02 Safe utilization rate of polluted farmland (0.0070)
	Ecological security (0.15)	A05 Ecological protection red line area (0.0084); B09 Area of city and rural construction land within the ecological protection red line (0.0088)
	Cultural security (0.14)	A06 Area of historical and cultural protection line (0.0069); B10 Natural and cultural heritage (0.0080); B11▲ Number of destroyed historical and cultural relics and environmental events (0.0064); T03 Proportion of citizens’ sense of historic landmark identity (0.0069); T04 Cultural recognition rate of citizens from Hehuang (0.0069)
	City resilience (0.15)	A07 Emergency shelter area per capita (0.0082); A08 Fire and rescue 5 min accessible coverage (0.0077); A09 City permeable surface proportion (0.0073); A10 Number of city waterlogging points (0.0069); A11 Number of super tall buildings (0.0069); B12▲ Proportion of integrated disaster reduction demonstration communities (0.0086); B13 Average annual land subsidence (0.0069); B14 The number of geological hazard points under control (0.0069); B15 Flood control dike compliance rate (0.0070); T05 Old pipe network renewal rate (0.0080); T06 Pipe corridor construction density (0.0070)
	Planning control (0.14)	A12 Number of illegal adjustments of planning, land use and sea use (0.0064)
	City security satisfaction (0.12)	S01 Public satisfaction with city security (0.0073)
Innovation (0.16)	Input−output (0.25)	B16▲ Social labor productivity (0.0070); B17 Research and experimental development investment intensity (0.0076); B18 Numbers of invention patents owned by 10,000 people (0.0079); B19 Number of institutions of higher Learning (0.0070); B20 Population with a university degree per 100,000 population (0.0079)
	Development model (0.25)	A13 Idle land disposal rate (0.0086); A14 Stock land supply ratio (0.0081); B21▲ Allotment and unsupplied land disposal rate (0.0086); B22▲ New area for city renewal and reconstruction (0.0089); A15 Proportion of city and rural industrial land in city and rural construction land (0.0074); B23▲ Ratio of city and rural residential land to city and rural construction land (0.0069); B24▲ City and rural land-use ratio (0.0064); A16 Proportion of land transfer revenue in government budget revenue (0.0069); B25▲ Comprehensive land price for city construction land (0.0071); A17 City road network density (0.0072)
	Wisdom city (0.25)	A18 Proportion of county-level units in the construction and application of “Unified Platform” (0.0064)
	City innovation satisfaction (0.25)	S02 Public satisfaction with city innovation (0.0069)
Coordination (0.15)	Agglomeration intensive (0.25)	A19 Permanent population (0.0074); B26▲ Actual service management population (0.0075); B27▲ Natural population growth rate (0.0075); B28▲ Permanent resident urbanization rate (0.0075); A20 City permanent population density (0.0072); A21 Total construction land area (0.0076); A22 City and rural construction land area (0.0080); B29 Area of city and rural construction land within the city development boundary (0.0078); B30 △ Intensity of land development (0.0071); A23 City building density (0.0070); T07 Proportion of permanent ethnic minority population (0.0077)
	City–rural integration (0.3)	B31▲ Per capita city construction land area (0.0070); B32▲ Per capita city residential land area (0.0084); B33▲ Per capita village construction land area (0.0070); B34▲ Level of hospital transportation 30 min coverage of administrative village (0.0079); B35 Highway access rate of administrative village level (0.0085); B36 Rural tap water penetration rate (0.0075); B37 Per capita disposable income ratio of city and rural residents (0.0064)
	Aboveground and underground coordination (0.25)	B41▲ Per capita underground space area (0.0080)
	City coordination satisfaction (0.20)	S03 Satisfaction with public city coordination (0.0069)
Green (0.20)	Ecological protection (0.25)	A24 Forest coverage rate (0.0073); A25 Forest stock (0.0069); B42 Forestland holdings (0.0069); B43 grassland area (0.0070); B44▲ Newly added ecological restoration area (0.0086); B45 Protection rate of key biological species (0.0074)
	Green and low-carbon production (0.25)	A26 Land consumption per ten thousand yuan GDP (0.0077); B46 Water consumption per ten thousand yuan GDP (0.0070); B47 Energy consumption per ten thousand yuan GDP (0.0069); B48 Carbon dioxide emission reduction per unit of GDP (0.0069); B49 Proportion of new energy and renewable energy (0.0069); B50▲ Average value added of industrial land (0.0077); B51 Integrated pipe gallery length (0.0076)
	Green and low-carbon life (0.25)	B52▲ City household waste recycling rate (0.0069); B53▲ Rural household waste treatment rate (0.0070); B54▲ Proportion of green transport trips (0.0073); B55 Proportion of green buildings in new and rebuilt buildings (0.0072)
	City green satisfaction (0.25)	S04 Public satisfaction with city green space (0.0069)
Openness (0.15)	Network connectivity (0.25)	B56 Number of cities with regular international flights (0.0070); B57 Number of cities with regular domestic flights (0.007); B58 Proportion of county units 1 h from central city international airport or mainline airport (0.0073)
	Foreign communication (0.25)	A27 Daily number of external people flow (0.0076); B59▲ Railway passenger volume (0.0069); B60 Airport annual passenger throughput (0.0071); B61 Annual number of domestic tourists (0.0071); B62 Number of inbound tourists (0.0069); B63 Number of international conferences, exhibitions and sports events (0.0071)
	Foreign trade (0.25)	B64 Annual cargo throughput (0.0071); B66 Total foreign trade volume (0.0070)
	Satisfaction with city opening (0.25)	S05 Public satisfaction with city opening (0.0069)
Sharing (0.16)	Appropriate industry (0.20)	B67 Number of new city jobs (0.0086); A28 Average weekday commuting time (0.0070); B68▲ Residents within 45 min commuting time (0.0078); B69 One-hour population coverage in metropolitan areas (0.0073); T08 Bus stop coverage (0.0064)
	Appropriate livable (0.20)	A29 15 min coverage of community life circle (0.0071); B71▲ 15 min walking coverage of community health service facilities (0.0070); A30 Number of beds in health institutions per 1000 residents (0.0082); B72▲ City-level hospital coverage 2 km (0.0077); B73▲ Number of kindergarten classes per 10,000 people (0.0074); B74▲ 10 min walking coverage of community primary schools (0.0084); B75▲ Community middle school 15 min walk coverage (0.0080); A31 Number of nursing beds per 1000 elderly people (0.0069); B76▲ 5 min walking coverage of community elderly care facilities (0.0084); B77▲ Burial land area (0.0069); B78▲ Community cultural activity facilities 15 min walking coverage (0.0077); B79▲ Wet market (fresh supermarket) 10 min walk coverage (0.0080); A32 Per capita city housing area (0.0073); B80▲ The proportion of new policy housing (0.0089); B81▲ Number of public rental housing units (0.0078); T09 City street cleanliness rate (0.0072); T10 Fine particulate matter (PM2.5) average annual concentration (0.0077); T11 PM10 concentration (0.0081); Overall noise compliance rate of T12 functional area (0.0072)
	Appropriate music (0.20)	B82▲ 15 min walking coverage of community sports facilities (0.0064); B83 Football ground facilities within 15 min; B84▲ Number of museums, libraries, science and technology museums, art galleries and other cultural and artistic venues per 100,000 people (0.0076); B85 Number of cafes and tea houses per 10,000 people (0.0070)
	Appropriate swimming (0.20)	B86▲ Park green space, square walking coverage of 5 min (0.0085); A33 Per capita park green space (0.0073); B87▲ Per capita greenway length (0.0077); B88 Forest walking coverage of 15 min (0.0070); B89 Number of days with good air quality (0.0071)
	City sharing satisfaction (0.20)	S06 Public satisfaction with city sharing (0.0069)

Notes: the values in parentheses are the weight values.

## Data Availability

The data presented in this study are available on request from the corresponding author. The data are not publicly available because some of the contents are classified.

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
