# Peer review of "City Health Examination and Evaluation of Territory Spatial Planning for SDG11 in China: A Case Study of Xining City in Qinghai Province"

_ijerph, 2023, doi:10.3390/ijerph20043243_

Round 1
Reviewer 1 Report
The article „City Health Examination and Evaluation of Territory Spatial Planning for SDG11 in China: A Case Study of Xining city in Qinghai Province” deals with a very important topic related to the quality of life in the city. As the number of people living in cities around the world is increasing, such a study could make an important contribution to assessing the quality of urban life, including the health of residents. The authors, using the TOPSIS method, indicated numerous additional indicators that should be taken into account in the city health examination, and the results of territorial spatial planning in Xining city. As the authors mentioned, such a study can be beneficial for enriching the methodology, adding appropriate indicators can be a significant help for city managers.
Although the topic addressed in the article is important and the proposed method could be applied to urban management, there are several concerns and comments that should be considered or included in the text. What is most questionable are the values of the indicators. The values of indicators determined by the authors without providing the basis for such an assessment is questionable. Even though the Discussion section mentions that the index with indicators “needs to be further refined and improved in future studies”, the article should explain where the values ​​indicated in the table come from. In Table 1 City health examination and evaluation index system of territorial spatial planning in Xining city", 6 indicators are listed in the Target layer column. Their values are given in parentheses. On what basis were the values assigned to the indicators? Similarly, the problem applies to the Rule layer column and the Index layer column. It would be necessary to indicate in the article the source of the values of each indicator. In the methodology (233), the authors describe in very general terms the origin of the values of each indicator and point to the literature where detailed information can be found. Similarly, the case applies to the TOPSIS method. It would be appropriate to briefly describe how the indicators were evaluated and also describe what the TOPSIS method is all about.
In addition:
1. In the line number 144 one should consider whether it is necessary to specify the location of the city in such detail - using coordinates? Is it not enough to determine the location of the city of Xining using administrative data or on a map?
2. Taking into account the numbering of the subsections concerning the results of individual indicators, in line number 431 it should be 2.2.6. Openness dimensions, and in line 454 – 2.2.7. Sharing dimensions.
Author Response
Response to Reviewer 1 Comments
Thank you very much for your valuable comments on the revision of this article. Revised portion are marked in red in the paper. The details are as follows:
Point 1: What is most questionable are the values of the indicators. The values of indicators determined by the authors without providing the basis for such an assessment is questionable. Even though the Discussion section mentions that the index with indicators “needs to be further refined and improved in future studies”, the article should explain where the values ​​indicated in the table come from.
Response 1: Dear reviewer, thank you very much for your valuable comments of our manuscript. I would like to make a statement about your comments. The social statistics data are mainly obtained by Statistical Yearbook of Xining City and the Xining Natural Resources and Planning Bureau from relevant commissions and bureaus through correspondence. Therefore, the values of indicators are credible and scientific. See in lines 192 in our revised manuscript.
Point 2: In Table 1 City health examination and evaluation index system of territorial spatial planning in Xining city", 6 indicators are listed in the Target layer column. Their values are given in parentheses. On what basis were the values assigned to the indicators? Similarly, the problem applies to the Rule layer column and the Index layer column. It would be necessary to indicate in the article the source of the values of each indicator.
Response 2: Dear reviewer, thank you very much for your valuable comments of our manuscript. I'm sorry that I didn't make it clear in our manuscript that the values in parentheses are the weight values not the index value. To ensure the scientific rigor of the index weight, the weight of each index is obtained by the entropy method, and the weights of the criterion layer and target layer are obtained by the combination of the entropy method and expert scoring method. A note has been added below Table 1(Notes: the values in parentheses are the weight values). See in lines 245 in our revised manuscript.
Point 3: In the methodology (233), the authors describe in very general terms the origin of the values of each indicator and point to the literature where detailed information can be found. Similarly, the case applies to the TOPSIS method. It would be appropriate to briefly describe how the indicators were evaluated and also describe what the TOPSIS method is all about.
Response 3: Dear reviewer, thank you very much for your comments, those comments are all valuable and very helpful for revising and improving our paper. I am very sorry that due to the limitation of the journal on the number words of articles, we removed the detailed introduction of the method in our original manuscript. According to your comments, we have made a supplement to the method used in our manuscript. See in lines 251-299 in our revised manuscript. See below for details:
2.4.1. Maximum–minimum method
The maximum–minimum values were used as the standard treatment for each index. The specific operation process is as follows:
- The Maximum–minimum method
According to the upper limit) and the lower limit thresholds of each indicator, the extreme value method is adopted to conduct dimensionless standardization of each indicator. The calculation formula is as follows:
For positive indicators: (1)
For negative indicators: (2)
In equations (1) and (2), is the standardized value of the extreme value method of the ith index in the jth year, andare the maximum and minimum values of the ith index respectively. Since the target value or optimal threshold of each indicator is not given in the Regulations, the ideal value obtained from various specifications, documents or plans is taken as the maximum value in this study.
2.4.2. Improved TOPSIS
The TOPSIS method is a multiobjective decision analysis method based on the proximity between the evaluation object and the ideal scheme [44]. The improved TOPSIS method is a multischeme decision-making method combined with the entropy method and the traditional TOPSIS method, which has the advantages of no strict restrictions on the data distribution and the number of indicators, simple calculation, and longitudinal comparison of the evaluation results [43]. The specific operation process is as follows:
â‘ The normalization matrix is constructed to obtain the dimensionless decision matrix
In equations (3), is the normalized value after dimensionless normalization, and is the normalized value of extreme value method. This study takes Xining City of Qinghai Province as the evaluation unit, selects 135 indicators in 2020 to evaluate the physical examination of Xining's territorial space planning, so m=2020, n=135.
â‘¡ Construct the normalized weighted decision matrix.
In equations (4), is the weight value of the ith index. is the normalized value after non-dimensionalization.
â‘¢ Determine the positive and negative ideal value schemes and .
In equations (5), is the maximum set of the ith index in the jth year; is the set of minimum values in the jth year of the ith index.
â‘£ Using weighted Euclidean distance measure distance .
In equations (6), and are the positive ideal value distance and negative ideal value distance of the jth year respectively.
⑤ Measure criterion layer city health index value .
In equations (7), is the urban health index value of the jth year criterion layer; is the weight of the criterion layer.
â‘¥ Measure the total value of urban health index.
In equations (8), and H are the value and total value of urban health index of sub-target layer in the jth year respectively. is the weight of the sub-target layer. H=1, the city health state is the best, vice versa, the city health state is the worst.
Point 4: In the line number 144 one should consider whether it is necessary to specify the location of the city in such detail - using coordinates? Is it not enough to determine the location of the city of Xining using administrative data or on a map?
Response 4: Dear reviewer, thank you very much for your valuable comments of our manuscript. I am very sorry that due to the limitation of the journal on the number words of articles, we omitted the location map, which caused trouble for readers. Therefore, according to the suggestions of you and reviewer 2, we add the location map of Xining City in line 167 in our revised manuscript, which is helpful to improve the readability of our manuscript. The location map is as follows:
Figure 1. The location map of study area (draw by authors)
At the same time, due to the addition of new pictures, I also changed the picture number of corresponding pictures later.
Point 5: Taking into account the numbering of the subsections concerning the results of individual indicators, in line number 431 it should be 2.2.6. Openness dimensions, and in line 454 – 2.2.7. Sharing dimensions.
Response 5: Dear reviewer, thank you very much for your valuable comments of our manuscript. I am very sorry that there may be a system error when we uploaded the manuscript according to the journal. We have corrected the wrong serial number. Line 431 should be 2.2.6, and 454 has been modified to 2.2.7.
Thanks again for your valuable comments!

Reviewer 2 Report
The article addresses current challenges in the field of "spatial planning for a healthy living environment". I consider it to be coherent in content, but make the following comments and suggestions for additions:
Lines: 13 - 34: The abstract is a bit too long, exceeds the rules of the journal (max. 200 words), especially in the first part, the text gives too long descriptions of commonly known facts...
Lines 63 - 64: "...establishing a national spatial planning system..." this statement is written as if spatial planning has not existed until now, are you sure about that? It would be appropriate to explain the current situation in the field of spatial planning in urban areas and in the rural hinterland.
Line 141: at the end of the introduction, what exactly is the research question(s)? It is necessary to define it more precisely.
Lines 144-145: It is unusual in the text of the article to give coordinates, but if you do, it would be appropriate to show this area on some kind of map.
Line 218: You should be aware that the table with the specified indexes is your work(?), so it should be given elsewhere, not here at the beginning. In general, you should first explain (in order!) the methodology of the work (for the whole article) and only then the detailed parameters (like these indexes)
Line 223: The explanation of the methodology should refer to the research questions, in the sense: what and how did you do to answer the questions asked?
Line 261: Figure 1: The sources must be indicated (authorship of this graphic or its components!)
Line 271: Interpretation of this graph: did the period of the epidemic (Covid-19) have an impact on it? Please explain if this is possible - based on the available data.
Line 515: In the conclusion, authors may express their opinion on key stakeholders, especially decision makers who play a major role in the spatial planning process. Such an "appeal" can effectively address specific institutions and their administrative steps to further improve the situation
Author Response
Response to Reviewer 2 Comments
Thank you very much for your valuable comments on the revision of this article. Revised portion are marked in red in the paper. The details are as follows:
Point 1: Lines: 13 - 34: The abstract is a bit too long, exceeds the rules of the journal (max. 200 words), especially in the first part, the text gives too long descriptions of commonly known facts...
Response 1: Dear reviewer, thank you very much for your valuable comments of our manuscript. According to your suggestions, we have condensed the abstract to 204 words. See in lines 14-35 in our revised manuscript and amended it as follows:
Abstract: City health examination and evaluation of territorial spatial planning is a new policy tool in China. However, research on city health examination and evaluation of territorial spatial planning is still in the exploratory stage in China. Guided by sustainable cities and communities (SDG11), a reasonable city health examination and evaluation index system of Xining city in Qinghai Province is constructed in this paper. The improved technique for order preference by similarity to ideal solution (TOPSIS) was used to quantify the evaluation results, and the city health index was visualized by the city health examination signals and warning panel. The results show that the city health index of Xining city continuously raised from 35.76 in 2018 to 69.76 in 2020. However, it is still necessary to address the shortcomings in innovation, coordination, openness and sharing and to improve the level of city space governance in a holistic way. This study is an exploration of the methodology used in city health examination and the evaluation of territorial spatial planning in China, which can provide a foundation for the sustainable development of Xining city and also provide a case reference for other cities seeking to carry out city health examinations and evaluations of territorial spatial planning in China.
Point 2: Lines 63 - 64: "...establishing a national spatial planning system..." this statement is written as if spatial planning has not existed until now, are you sure about that? It would be appropriate to explain the current situation in the field of spatial planning in urban areas and in the rural hinterland.
Response 2: Dear reviewer, thank you very much for your valuable comments of our manuscript. After the reform of the “large ministry system” in 2018, the responsibility of planning preparation was transferred to the Ministry of Natural Resources of the People’s Republic of China. With the promotion of "integration of multiple plans", “The Several Opinions of the CPC Central Committee and the State Council on Establishing a Territorial Space Planning System and Supervising Its Implementation” issued by the CPC Central Committee and the State Council further clarify the need to establish a territorial space planning system and supervise its implementation. So the territorial space planning system and supervise its implementation are in the exploratory stage, and the city health evaluation is also an important measure for establishing a national spatial planning system and supervising the implementation of territorial spatial planning in China.
Point 3: Line 141: at the end of the introduction, what exactly is the research question(s)? It is necessary to define it more precisely.
Response 3: Dear reviewer, thank you very much for your valuable comments of our manuscript. According to your suggestions, we have clarified our research questions at the end of the introduction. Our research questions are how to construct a reasonable city health examination and evaluation index system of territorial spatial planning and quantify and visualize the evaluation results are very important, here, we attempt to construct a reasonable city health examination and evaluation index system of territorial spatial planning, and the improved TOPSIS method was applied to quantify and visualize the evaluation results for better understanding the sustainable cities and communities. See in lines 134-139 in our revised manuscript.
Point 4: Lines 144-145: It is unusual in the text of the article to give coordinates, but if you do, it would be appropriate to show this area on some kind of map.
Response 4: Dear reviewer, thank you very much for your valuable comments of our manuscript. I am very sorry that due to the limitation of the journal on the number words of articles, we omitted the location map, which caused troubles for readers. Therefore, according to the suggestion of you and reviewer 1, we add the location map of Xining City in line 167 in our revised manuscript, which is helpful to improve the readability of our manuscript. The location map is as follows:
Figure 1.The location map of study area (draw by authors)
At the same time, due to the addition of new pictures, I also changed the picture number of corresponding pictures later.
Point 5: Line 218: You should be aware that the table with the specified indexes is your work(?), so it should be given elsewhere, not here at the beginning. In general, you should first explain (in order!) the methodology of the work (for the whole article) and only then the detailed parameters (like these indexes)
Response 5: Dear reviewer, thank you very much for your valuable comments of our manuscript. Yes, the table with the specified indexes is my work. As shown in Figure 2, because we first constructed the index system, and then chose the appropriate methodology.Hence the structure of our manuscript are first explain the table with the specified indexes, and then explain the methodology of our work. Certainly, your suggestion is also of great help for us, in orde to make it easier for readers to understand our work, we add a framework description of the revised manuscript at the end of the introduction. See in lines 155-162 in our revised manuscript.
Figure 2. The construction process diagram of the city health examination signal light and warning panel in territorial spatial planning. (draw by authors )
The rest of this paper is organized as follows. Section 2 introduces the study area and data sources which includes the social statistics data and natural resource data. And introduces the index system and research methods in detail. As the core section, the Section 3 presents the results and analysis in which we analyzed the characteristics of the six dimensions of the city health index. Section 4 provides a discussion comprised of the rationality of the improved TOPSIS method, suggestions on city sustainable development and community construction, limitations and prospects. The last section presents our conclusions.
Point 6: Line 223: The explanation of the methodology should refer to the research questions, in the sense: what and how did you do to answer the questions asked?
Response 6: Dear reviewer, thank you very much for your valuable comments of our manuscript. I am very sorry that due to the limitation of the journal on the number words of articles, we removed the detailed introduction of the research method. According to your comments, we have made a supplement to the method used in our manuscript. See in lines 251-299 in our revised manuscript. See below for details:
2.4.1. Maximum–minimum method
The maximum–minimum values were used as the standard treatment for each index. The specific operation process is as follows:
- The Maximum–minimum method
According to the upper limit) and the lower limit thresholds of each indicator, the extreme value method is adopted to conduct dimensionless standardization of each indicator. The calculation formula is as follows:
For positive indicators: (1)
For negative indicators: (2)
In equations (1) and (2), is the standardized value of the extreme value method of the ith index in the jth year, andare the maximum and minimum values of the ith index respectively. Since the target value or optimal threshold of each indicator is not given in the Regulations, the ideal value obtained from various specifications, documents or plans is taken as the maximum value in this study.
2.4.2. Improved TOPSIS
The TOPSIS method is a multiobjective decision analysis method based on the proximity between the evaluation object and the ideal scheme [44]. The improved TOPSIS method is a multischeme decision-making method combined with the entropy method and the traditional TOPSIS method, which has the advantages of no strict restrictions on the data distribution and the number of indicators, simple calculation, and longitudinal comparison of the evaluation results [43]. The specific operation process is as follows:
â‘ The normalization matrix is constructed to obtain the dimensionless decision matrix
In equations (3), is the normalized value after dimensionless normalization, and is the normalized value of extreme value method. This study takes Xining City of Qinghai Province as the evaluation unit, selects 135 indicators in 2020 to evaluate the physical examination of Xining's territorial space planning, so m=2020, n=135.
â‘¡ Construct the normalized weighted decision matrix.
In equations (4), is the weight value of the ith index. is the normalized value after non-dimensionalization.
â‘¢ Determine the positive and negative ideal value schemes and .
In equations (5), is the maximum set of the ith index in the jth year; is the set of minimum values in the jth year of the ith index.
â‘£ Using weighted Euclidean distance measure distance .
In equations (6), and are the positive ideal value distance and negative ideal value distance of the jth year respectively.
⑤ Measure criterion layer city health index value .
In equations (7), is the urban health index value of the jth year criterion layer; is the weight of the criterion layer.
â‘¥ Measure the total value of urban health index.
In equations (8), and H are the value and total value of urban health index of sub-target layer in the jth year respectively. is the weight of the sub-target layer. H=1, the city health state is the best, vice versa, the city health state is the worst.
Point 7: Line 261: Figure 1: The sources must be indicated (authorship of this graphic or its components!)
Response 7: Dear reviewer, thank you very much for your valuable comments of our manuscript. According to your suggestions, we add the sources of Figure 2. See below for details:
Figure 2. The construction process diagram of the city health examination signal light and warning panel in territorial spatial planning. ( draw by authors)
Point 8: Line 271: Interpretation of this graph: did the period of the epidemic (Covid-19) have an impact on it? Please explain if this is possible - based on the available data.
Response 8: Dear reviewer, thank you very much for your valuable comments of our manuscript. It is true that the COVID-19 epidemic has had a negative impact on the economic development and urbanization process of the Xining City, but we lack relevant statistical datas, so we have qualitatively analyzed the impact of the COVID-19 epidemic in our results. Especially in terms of openness, the impact of the COVID-19 epidemic is the most significant, we analyzed the impact of the COVID-19 epidemic. See in lines 344-345 and 526-527 in our revised manuscript.
Point 9: Line 515: In the conclusion, authors may express their opinion on key stakeholders, especially decision makers who play a major role in the spatial planning process. Such an "appeal" can effectively address specific institutions and their administrative steps to further improve the situation.
Response 9: Dear reviewer, thank you very much for your valuable comments of our manuscript. Yes, the decision makers are very important in the process of preparation of territorial spatial planning, thank you very much for acknowledging our "appeal". And in our discussion, we put forward corresponding suggestions for urban spatial governance and sustainable development of Xining from the perspective of decision makers in six dimensions, so that to provide some help and enlightenment for decision makers. See in lines 552-592 in our revised manuscript.
Thanks again for your valuable comments!

Reviewer 3 Report
The paper is relevant and well written. Just tiny revision.
Your abstract seems well organized, but the results part in the abstract lacks data, not recommended to use text only.
General comment on the Introduction section: My main suggestion is to shorten the introduction that is a bit too long and to make a deeper analysis of the most recent literature. Besides, some knowledge and methodological backgrounds were not presented in the introduction and methodology but with results.
As for Table 1, this part needs to focus on the specific research content. At present, the length of Table 1 needs to be moderately reduced.
The research method is too simple and needs to be elaborated in detail. Specifically, statistical research methods related to the improved TOPSIS method need further explanation.
Conclusions: Focus on your results/findings.
The format of the paper needs to be modified according to the template of the journal.
I have no strong plagiarism checker and you should do that.
Author Response
Response to Reviewer 3 Comments
Thank you very much for your valuable comments on the revision of this article. Revised portion are marked in red in the paper. The details are as follows:
Point 1: Your abstract seems well organized, but the results part in the abstract lacks data, not recommended to use text only.
Response 1: Dear reviewer, thank you very much for your valuable comments of our manuscript. According to your suggestions, we have condensed the abstract to 204 words. See in lines 14-35 in our revised manuscript and amended it as follows:
Abstract: City health examination and evaluation of territorial spatial planning is a new policy tool in China. However, research on city health examination and evaluation of territorial spatial planning is still in the exploratory stage in China. Guided by sustainable cities and communities (SDG11), a reasonable city health examination and evaluation index system of Xining city in Qinghai Province is constructed in this paper. The improved technique for order preference by similarity to ideal solution (TOPSIS) was used to quantify the evaluation results, and the city health index was visualized by the city health examination signals and warning panel. The results show that the city health index of Xining city continuously raised from 35.76 in 2018 to 69.76 in 2020. However, it is still necessary to address the shortcomings in innovation, coordination, openness and sharing and to improve the level of city space governance in a holistic way. This study is an exploration of the methodology used in city health examination and the evaluation of territorial spatial planning in China, which can provide a foundation for the sustainable development of Xining city and also provide a case reference for other cities seeking to carry out city health examinations and evaluations of territorial spatial planning in China.
Point 2: General comment on the Introduction section: My main suggestion is to shorten the introduction that is a bit too long and to make a deeper analysis of the most recent literature.
Response 2: Dear reviewer, thank you very much for your valuable comments of our manuscript. According to your suggestion, we have simplified the introduction and added new literature. See in lines 69-154 in our revised manuscript.
Point 3: Besides, some knowledge and methodological backgrounds were not presented in the introduction and methodology but with results. The research method is too simple and needs to be elaborated in detail. Specifically, statistical research methods related to the improved TOPSIS method need further explanation.
Response 3: Dear reviewer, thank you very much for your comments, those comments are all valuable and very helpful for revising and improving our paper. I am very sorry that due to the limitation of the journal on the number words of articles, we removed the detailed introduction of the method in our original manuscript. According to your comments, we have made a supplement to the method used in our manuscript. See in lines 251-299 in our revised manuscript. See below for details:
2.4.1. Maximum–minimum method
The maximum–minimum values were used as the standard treatment for each index. The specific operation process is as follows:
- The Maximum–minimum method
According to the upper limit) and the lower limit thresholds of each indicator, the extreme value method is adopted to conduct dimensionless standardization of each indicator. The calculation formula is as follows:
For positive indicators: (1)
For negative indicators: (2)
In equations (1) and (2), is the standardized value of the extreme value method of the ith index in the jth year, andare the maximum and minimum values of the ith index respectively. Since the target value or optimal threshold of each indicator is not given in the Regulations, the ideal value obtained from various specifications, documents or plans is taken as the maximum value in this study.
2.4.2. Improved TOPSIS
The TOPSIS method is a multiobjective decision analysis method based on the proximity between the evaluation object and the ideal scheme [44]. The improved TOPSIS method is a multischeme decision-making method combined with the entropy method and the traditional TOPSIS method, which has the advantages of no strict restrictions on the data distribution and the number of indicators, simple calculation, and longitudinal comparison of the evaluation results [43]. The specific operation process is as follows:
â‘ The normalization matrix is constructed to obtain the dimensionless decision matrix
In equations (3), is the normalized value after dimensionless normalization, and is the normalized value of extreme value method. This study takes Xining City of Qinghai Province as the evaluation unit, selects 135 indicators in 2020 to evaluate the physical examination of Xining's territorial space planning, so m=2020, n=135.
â‘¡ Construct the normalized weighted decision matrix.
In equations (4), is the weight value of the ith index. is the normalized value after non-dimensionalization.
â‘¢ Determine the positive and negative ideal value schemes and .
In equations (5), is the maximum set of the ith index in the jth year; is the set of minimum values in the jth year of the ith index.
â‘£ Using weighted Euclidean distance measure distance .
In equations (6), and are the positive ideal value distance and negative ideal value distance of the jth year respectively.
⑤ Measure criterion layer city health index value .
In equations (7), is the urban health index value of the jth year criterion layer; is the weight of the criterion layer.
â‘¥ Measure the total value of urban health index.
In equations (8), and H are the value and total value of urban health index of sub-target layer in the jth year respectively. is the weight of the sub-target layer. H=1, the city health state is the best, vice versa, the city health state is the worst.
Point 4: As for Table 1, this part needs to focus on the specific research content. At present, the length of Table 1 needs to be moderately reduced.
Response 4: Dear reviewer, thank you very much for your valuable comments of our manuscript. Because we according to the "Regulations for Healthy Examination and Evaluation of Territory Spatial Planning" (TD/T1063-2021), and combined with the “Beautiful China Construction Evaluation Index System and Implementation Plan” and “Guidelines for the Formulation of Municipal Territorial Space Master Plan (Tri-al)”, we constructed the index system of our work, so there are many indicators to evaluate the city healthy, and it is the most simplified state at present.
Point 5: Conclusions: Focus on your results/findings.
Response 5: Dear reviewer, thank you very much for your valuable comments of our manuscript. According to your suggestions, we have refined our conclusions, based on our findings. See in lines 598-607 in our revised manuscript.
Point 6: The format of the paper needs to be modified according to the template of the journal.
Response 6: Dear reviewer, thank you very much for your valuable comments of our manuscript. The format of our manuscript has been modified according to the template of the journal.
Point 7: I have no strong plagiarism checker and you should do that.
Response 7: Dear reviewer, thank you very much for your valuable comments of our manuscript. Follow the journal's submission requirements, journals have been monitored for academic misconduct before the peer review. Therefore, we can guarantee the originality of the article and there is no plagiarism.
Thanks again for your valuable comments!
